# Numerical Simulation and Process Optimization of Internal Thread Cold Extrusion Process

**DOI:** 10.3390/ma13183960

**Published:** 2020-09-07

**Authors:** Hong-Ling Hou, Guang-Peng Zhang, Chen Xin, Yong-Qiang Zhao

**Affiliations:** 1School of Mechanical and Precision Instrumental Engineering, Xi’an University of Technology, Xi′an 710048, China; xjtuhhl@163.com; 2School of Mechanical Engineering, Shaanxi University of Technology, Hanzhong 723001, China; 15565520069@163.com (C.X.); zyq0620@163.com (Y.-Q.Z.)

**Keywords:** internal thread cold extrusion, numerical simulation, physical test, process parameter optimization

## Abstract

In the internal thread extrusion forming, if the process parameters are not selected properly, the extrusion torque will increase, the extrusion temperature will be too high, or even the tap will break. In order to obtain effective process parameters under certain working conditions, this paper uses a combination of numerical simulation and process experiment to analyze the influence of the bottom hole diameter, extrusion speed, and friction factor on the extrusion torque and extrusion temperature. Through an orthogonal experiment, the significant influence law of different process parameters on the extrusion torque and extrusion temperature was studied, and the order of their influence was determined. Based on the optimal process parameters, numerical simulations and process tests were carried out, and the extrusion effect and related parameters were compared and analyzed. The results show that the extruded thread has clear contour, uniform tooth pitch, complete tooth shape, and good flatness. Compared with before optimization, the maximum extrusion torque has been reduced by 37.15%, the maximum temperature has been reduced by 29.72%, and the extrusion quality has been improved. It shows that the optimized method and optimized process parameters have good engineering practicability.

## 1. Introduction

In recent years, China has vigorously developed the equipment manufacturing industry, such as rail transit equipment, automobiles and parts, machine tool manufacturing, robots, etc., and it has developed many new equipment and machines, among which threaded connections are indispensable; its performance is directly related to the service life of related equipment. Almost 100% of the bolts on high-end equipment in China are imported, such as special bolts for excavators, bolts for engine crankshafts, and bolts for aircraft landing gear. To improve the reliability and service life of threaded connections, more and more scholars are paying attention to the forming process and forming methods of threads, and propose to use extrusion forming technology to improve the strength, hardness, and fatigue resistance of threads.

In the extrusion molding process, the metal fiber is not cut, the thread surface structure is fibrous and distributed along the tooth shape, and there is a certain hardened layer and residual compressive stress on the tooth root surface. Therefore, it has better comprehensive performance and better performance than machined threads, as well as a large carrying capacity [1,2]. Extrusion processing can effectively improve the strength and hardness of internal threads, extend their service life, and have high processing accuracy and good surface quality. Now, it has become the first choice for small-size thread processing in the electronics industry. However, when the internal thread is extruded, an improper selection of process parameters often results in an unqualified thread profile, sharp increase in tap torque, abnormal breakage, and an excessively high extrusion temperature, which restrict the development and application of internal thread extrusion technology to a certain extent, so the research and optimization of extrusion process parameters is very important to processing quality.

There are many factors that affect the extrusion process of internal threads, and scholars at home and abroad have conducted various studies on key extrusion technologies. Fan Shuqin [3] introduced and explored the key technologies affecting the cold extrusion process of internal threads; Miu Hong, Zhang Min [4,5,6,7,8,9,10], and others conducted experiments on Q460, 300 M high-strength steel, and they predicted the quality of internal thread based on BP (Back Propagation) neural network. Huang Xiaolong, Li Yongyi [11,12,13], and others initially used numerical simulation technology to optimize the internal thread extrusion process. Peter Monka, Fromentin [14,15], and others conducted experimental research on the failure and breaking of taps during the tapping process, and they analyzed the influence of tool geometry parameters on the tapping process and the impact on the thread surface quality; Kosarev [16,17] studied the measurement technology of the quality of extruded internal threads and proposed a control method to estimate the manufacturing accuracy of internal threads.

This article starts from reducing the extrusion torque and extrusion temperature to improve the quality of the extrusion thread and the life of the extrusion tap, using a combination of numerical simulation and experimental design. Then, we obtain the influence of processing parameters on the index value through range analysis. The optimal combination parameters of extruded internal threads were optimized and verified by experiments.

## 2. Mechanism of Internal Thread Cold Extrusion

Internal thread extrusion is different from traditional cutting and tapping. It is a metal plastic processing method that uses extrusion taps to form a tooth profile. When the tap extrudes the metal material, the plastic deformation of the metal is carried out in a limited space. The metal in the deformation zone is subjected to a cyclical squeezing force, and an internal thread is formed after several times of extrusion. As shown in Figure 1a, a hole of a certain size is prefabricated on the workpiece according to the size of the thread to be extruded. The axial feed and rotation of the tap are coordinated with each other and introduced by the hole. The teeth of the tap are in intermittent contact with the workpiece. The metal flows along the ridge of the tap, accumulating and increasing. When the tap leaves the workpiece, the squeezed part is quite unloaded, the elastic deformation is restored, the plastic deformation remains, and finally a thread profile is formed.

In order to reduce the frictional resistance between the tap and the workpiece, the cross-section of the tap is often made into a polygon, as shown in Figure 1b. During normal operation, only the edge of the tap is squeezing the workpiece. The higher the number of edges of the tap, the more stable the deformation of the metal during extrusion, but the contact area between the teeth and the workpiece increases and the extrusion torque increases, so the number of edges should be selected according to the actual situation. In this study, the tap is four-sided. In contrast, the extrusion process forms a hardened layer on the surface of the internal thread, which greatly improves the surface hardness and strength of the internal thread.

## 3. Model Building

### 3.1. Establishing the Geometric Model of Extrusion Taps

The cross-section of the extrusion tap is a special curved edge prism, which is formed by relief grinding on a special thread grinder. According to the relative movement between the tap blank and the grinding wheel during the relief grinding process, the tap parameter equation is established from the movement trajectory of the grinding wheel as follows
(1)$$\{\begin{array}{c} x=R_{m}cos(\beta -\omega )+\left[R_{m}+d-\frac{K}{2}\left(1-cosn\omega \right)\right]cos\omega \\ y=R_{m}sin(\beta -\omega )+\left[R_{m}+d-\frac{K}{2}\left(1-cosn\omega \right)\right]sin\omega \\ tg\beta =\frac{-Knsinn\omega }{2R_{m}+2d-K+Kcosn\omega } \end{array}$$
where *R_m_* is the radius of the grinding wheel (*R_m_* = 200 mm).

*β*—angle between a point on the grinding wheel and the horizontal direction;*ω*—rotation angle of the tap blank;*d*—major diameter of the tap;*K*—amount of shovel back; and*n*—number of tap edges.

Take the M8 × 1.25 mm internal thread as an example, select the four-sided edge, and establish the geometric model of the extrusion tap. The major diameter of the tap is calculated by Formula (2):(2)$$d=\left(d_{0}+0.102P\right)$$
where *d*_0_ is the nominal diameter and *P* is the pitch.

The maximum amount of shovel back k is
(3)$$K_{max}=\frac{2R_{S\;\;}\left(R_{m}+R_{S}\right)}{n^{2}R_{m}}$$
where *R*_s_ is the radius of the extrusion tap.

When the amount of shovel back is less than *K*_max_, a complete cross-sectional shape can be shoveled. After calculating, *K*_max_ is 0.51. The amount of shovel back in this design is approximated by Formula (4).
(4)$$K=0.04d_{0}=0.32<K_{max}$$

Take the guide cone angle as 60° and the calibration cone chamfer slope as 0.085°. The cross-sectional curve of the extrusion tap is obtained from the Formulas (1), (2), and (4), as shown in Figure 2. The geometric model of the extrusion tap is established through this curve as shown in Figure 3.

### 3.2. Establishment of Finite Element Model

(1) Material constitutive equation.

The thread extrusion process is a large plastic deformation, volume-forming process, finite element involves material nonlinearity, geometric nonlinearity, contact nonlinearity, and other problems, so the rigid-plastic finite element theory is used to solve it.

When modeling, set the extrusion tap as a rigid body, the material is W6Mo5Cr4V2 high-speed steel; the workpiece is a plastic body, the material is 45 steel, the size is Φ 30 × 10 mm. Refine the grid in the range of R3–R5 mm (material deformation zone) of the center of the workpiece, and set the number of grids to 100,000. The workpiece is set to be restrained in the three directions of X, Y, and Z, keeping the position fixed. The movement parameters of the extrusion tap are set according to the extrusion speed, and each lead is fed along the axial direction while rotating by 2π rad. Set the relevant parameters according to Table 1, select the Lagrangian incremental type, and use the direct iteration method for simulation. After the finite element simulation is over, the “torque” function can be selected in the “load displacement diagram” module built in DEFORM-3D software to obtain the change trend of the extrusion torque during the extrusion process. In the “Summary” module, select “Heat Transfer Mode: Temperature” to obtain the temperature trend during the extrusion process. The results are shown in Figure 4.

Figure 4a is the finite element model of the workpiece and the tap. Figure 4b is a diagram of the simulated internal thread extrusion process. Figure 4c is a working torque change diagram during the extrusion process.

It can be seen from Figure 4c that at the beginning of the extrusion, the teeth of the tap are in contact with the workpiece, and the metal undergoes greater plastic deformation, resulting in a sharp increase in the load of the tap; as the extrusion process progresses, the calibration part of the tap participates in the extrusion, the metal gradually fills the tooth slots of the tap, and this process is slightly slow; the tap continues to squeeze downward, and the extrusion load decreases due to the inverted taper angle until the entire thread extrusion is completed. Figure 4d shows the metal flow direction during the internal thread extrusion process. As the arrow points, the metal gathers from the bottom of the tooth to the top of the tooth.

## 4. Finite Element Analysis and Process Optimization

The goal of process optimization is to reduce the working load and improve forming quality. With certain extrusion equipment and tools, the extrusion quality mainly depends on the extrusion process. Furthermore, the extrusion process mainly depends on the selection and combination of process parameters. In the extrusion process, extrusion torque is a comprehensive index that reflects the difficulty of extrusion of internal threads. Excessive torque makes extrusion difficult and tools are easily damaged; the extrusion temperature affects the quality of the extrusion threads and the life of the extrusion taps. Find the law of the influence of multiple process parameters on torque and temperature. Through a combination of simulation and physical experiment, optimize the best parameter combination, and then use the optimized results to guide the experiment.

### 4.1. The Influence of Various Factors

(1) Diameter of bottom hole.

Before extruding the internal thread of M8, according to the experience, holes with diameters of 7.27 mm, 7.32 mm, 7.37 mm, and 7.42 mm were prefabricated on the four blanks, respectively, and the extrusion simulation under different diameters was carried out; the extrusion effect and work torque are shown in Figure 5. It can be seen that the diameter of the bottom hole determines the fullness of the tooth shape and the working torque. When the diameter of the bottom hole is relatively small, the tooth profile is full, but the torque is large after extrusion; when the diameter of the bottom hole is large, the tooth profile is not clear, but the torque is small.

(2) Extrusion speed.

In order to observe the influence of extrusion speed on working torque and extrusion temperature, four speeds of 15 r/min, 30 r/min, 45 r/min, and 60 r/min were selected for simulation, and the effect of extrusion speed on torque and workpiece temperature was obtained. The law of influence is shown in Figure 6.

It can be seen that when the internal thread is extruded at different extrusion speeds, the overall change trend of the extrusion torque remains unchanged, and both increase at first and then decrease. When the extrusion speed is changed from 15 to 60 r/min, the extrusion torque is increased from 8200 to 17,400 N·mm, but the extrusion time is significantly shortened and the efficiency is improved. It can also be seen that as the extrusion process progresses, the trend of temperature changes first increases and then decreases. When the extrusion speed changes from 15 to 60 r/min, the extrusion temperature increases from 57.6 to 96.2 °C; if the speed is too high, the temperature of the metal deformation zone of the workpiece will rise sharply. At this time, the tap and the internal thread surface are prone to adhesion, which will cause strain on the workpiece surface.

(3) Friction factor.

When the internal thread is extruded, strong friction occurs between the tap and the surface of the workpiece and the metal is plastically deformed, which will generate a lot of heat. Effectively cool and lubricate the extrusion deformation zone to reduce the friction between the workpiece and the tap, reduce the working torque of extrusion, and increase the service life of the extrusion tap. For solids, the sliding friction factor changes with the relative speed, the contact properties are different, and the specific functions are also different. This complex situation can be approximated as the Coulomb friction law. In this paper, combining the extrusion load and the characteristics of the material, after simplifying the calculation, the friction coefficients are respectively taken as 0.08, 0.12, 0.20, and 0.25 for simulation tests, the internal thread simulation extrusion was performed on 45# steel and aluminum alloy, respectively, and the change of working torque obtained is shown in Figure 7. It can be seen that under the same lubrication condition, different metal materials require different torques during extrusion. Therefore, during the process of internal thread extrusion, a suitable and effective lubricant must be selected; otherwise, the extrusion torque will increase.

### 4.2. Process Parameters Optimization

On the basis of analyzing the influence of various factors on the extrusion torque and temperature, the orthogonal test is used to further optimize the process parameters to obtain the optimal combination of parameters.

#### 4.2.1. Experimental Design

Select the bottom hole diameter, extrusion speed, and friction factor as indexes, and assume that these three indexes do not have any interaction. Use the three-factor four-level standard to carry out the orthogonal test design. The design factor level is shown in Table 2. A total of 16 sets of simulation tests were conducted to optimize the extrusion torque and extrusion temperature. In the finite element software, 16 groups of simulation tests were set up to calculate the corresponding torques and extrusion temperatures. The simulation results are shown in Table 3.

#### 4.2.2. Data Analysis and Optimization

Firstly, the range analysis is performed on the two targets according to the single-index analysis method, and the optimal combination of the two targets is selected; then, the two sets of optimal results are comprehensively analyzed to obtain the final optimal combination.

(1) Range analysis.

The range analysis of the extrusion torque and extrusion temperature in Table 3 is carried out. The calculation and analysis results are shown in Table 4, *K*_i_ represents the sum of the corresponding test results when the level number is i in any column (I = 1,2,3,4).$\;\bar{K_{i}}$ represents the average of *K*_i_.

According to the range value R in the simulation results, the order of the factors affecting the extrusion torque is A > C > B; that is, the bottom hole diameter has the greatest influence, the friction factor is a few times smaller, and the extrusion speed has the least influence. The order of various factors affecting the extrusion temperature is B > A > C—that is, extrusion speed > bottom hole diameter > friction factor. According to the corresponding value of each factor in Table 4, plot the influence trend of factors on torque and temperature, as shown in Figure 8a,b. It can be seen that when the extrusion torque is the target, the optimal process parameter combination is A4B4C1; when the extrusion temperature is the target, the optimal process parameter combination is A4B1C4.

(2) Comprehensive optimization combination

The above two optimized combination schemes are not exactly the same, the comprehensive balance method is used to further analyze and optimize the orthogonal test results. As shown in Figure 9, according to $\bar{K_{1}}$, $\bar{K_{2}}$, $\bar{K_{3}}$, and $\bar{K_{4}}$ in Table 3, we investigate the influence of three factors A, B, and C on the extrusion torque and extrusion temperature. For influencing factor A, selecting the A4 level can simultaneously ensure the smallest extrusion torque and the smallest extrusion temperature. In the same way, B1 and C1 can be obtained respectively as the optimal levels of the two evaluation indicators. Considering the influence of bottom hole diameter A, extrusion speed B, and friction factor C on the extrusion torque and extrusion temperature, the optimal test plan combination is finally obtained as A4B1C2 (that is, the diameter of the bottom hole of the workpiece D = 7.40 mm, the extrusion speed n = 30 r/min, and the friction factor f = 0.08).

## 5. Experimental Verification

### 5.1. Optimized Simulation Results

Guided by the optimized process parameter A4B1C2, the internal thread extrusion simulation was carried out and compared with the tooth profile, extrusion torque, and extrusion speed before optimization, as shown in Figure 10 (the process parameters before optimization are extrusion speed, 60 r/min; bottom hole diameter, 7.35 mm; friction coefficient, 0.20).

It can be seen that there is no significant change in the height of the internal thread before and after optimization, but the maximum torque is reduced from 25.3 to 15.9 N·m, which is a decrease of 37.15%; the maximum temperature is reduced from 210.3 to 147.8 °C, which is a decrease of 29.72%. Optimizing process parameters can greatly reduce the torque and extrusion temperature of the tap during processing, making the extrusion easier. In addition, the service life of the extrusion tap is greatly extended, and the processed surface of the workpiece is prevented from sticking to the tap due to high temperature, and the processing quality and processing accuracy are improved.

### 5.2. Extrusion Test Results

A test piece with a thickness of 10 mm was cut from a No. 45 steel bar with a diameter of 30 mm, and a 7.40 mm bottom hole was prepared on this test piece with a standard twist drill and a reamer through a drilling–reaming process. According to the optimized parameters and actual processing conditions, the physical extrusion test of M8 × 1.25 internal thread was carried out, and a complete extrusion sample was obtained. An enlarged view of the extruded tooth-shaped root, side, and top is shown in Figure 11.

It can be seen that the degree of extrusion and plastic deformation of the top, side, and root of the thread profile are different. The metal flow is uniform at the root and side of the thread, dense, and strengthened, and the extrusion quality is high; the metal flow of the tooth top is not obvious, there is a little defect, and the extrusion is relatively poor, but it does not affect the use of the thread.

Observing and analyzing the extrusion effects before and after optimization in Figure 12, it can be seen that the effects of the optimization process are significant. The thread profile is clear, the tooth pitch is uniform, the tooth shape is relatively complete, the flatness is good, there is no obvious trace, and the surface roughness is small.

Through the simulation tests, the tooth height is 0.839 mm before the optimization process and it is 0.823 mm after optimization, the ratios of the formed tooth height and the required tooth height are 77.47% and 76.04%, respectively. The simulation results are also verified with experimental tests. The experiment results are shown in Figure 12. The tooth height values are 0.852 mm before optimization and 0.840 mm after optimization, the ratios of the formed tooth height and required tooth height are 78.68% and 77.62%, respectively. The simulations results are in good agreement with the experiment results.

## 6. Conclusions

(1)In order to achieve the best effect of extrusion, orthogonal experiments are carried out with the bottom hole diameter, extrusion speed, and friction factor as factors, and the influence of these parameters on the extrusion torque and extrusion temperature is analyzed, and the optimal process is obtained. The parameter combination is A4B1C1; that is, the diameter of the bottom hole of the workpiece is 7.40 mm, the extrusion speed is 30 r/min, and the friction factor is 0.08.(2)The optimized parameters are used to carry out the internal thread extrusion forming test. The test piece is in good agreement with the simulated extrusion results and meets the requirements in terms of forming shape and accuracy, which further proves the effectiveness of the process method.

## Figures and Tables

**Figure 1 materials-13-03960-f001:**
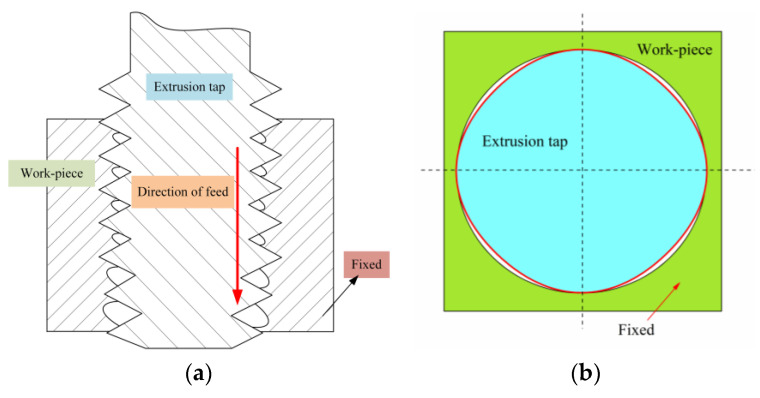
The principle of extrusion tapping in (**a**) and the working section of the tap in (**b**).

**Figure 2 materials-13-03960-f002:**
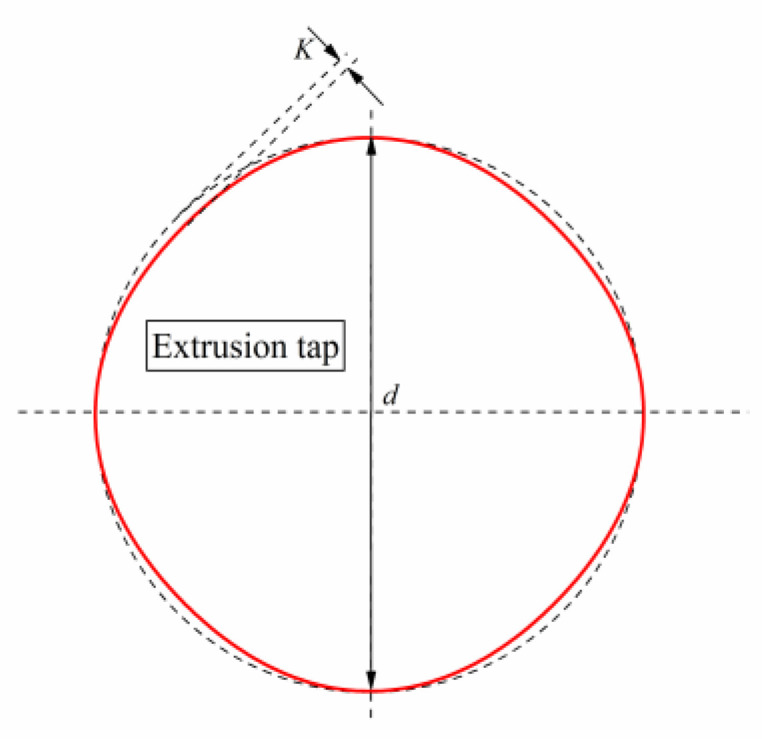
Extrusion tap profile line.

**Figure 3 materials-13-03960-f003:**
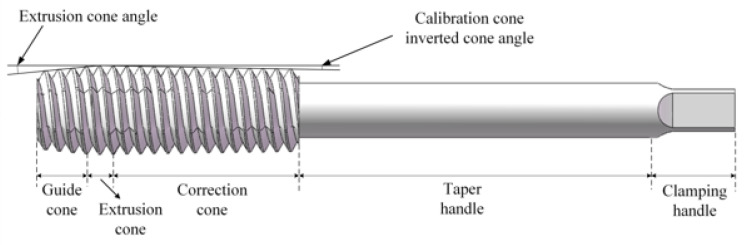
Extrusion tap geometry model.

**Figure 4 materials-13-03960-f004:**
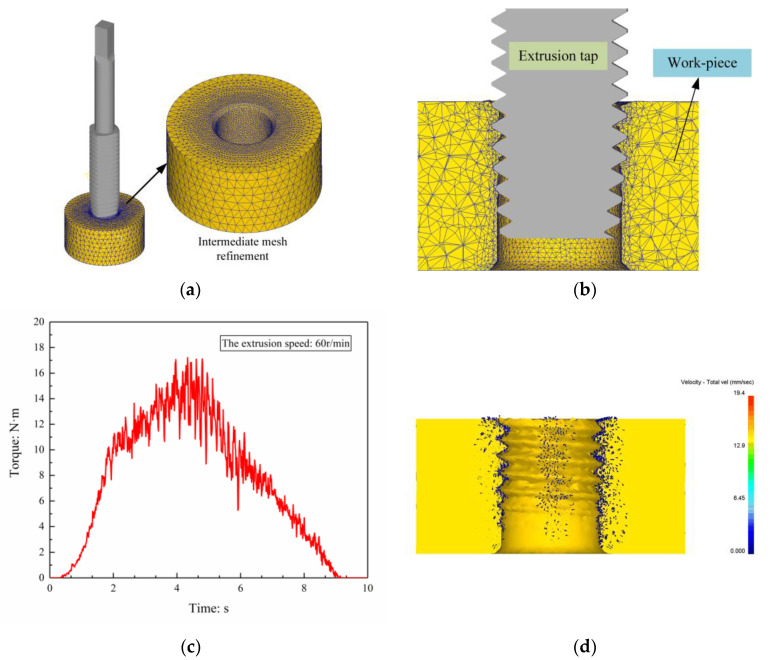
Simulation results of internal thread extrusion (**a**) FEM, (**b**) Extrusion process model, (**c**) Torque change curve during extrusion, (**d**) Metal flow direction during extrusion.

**Figure 5 materials-13-03960-f005:**
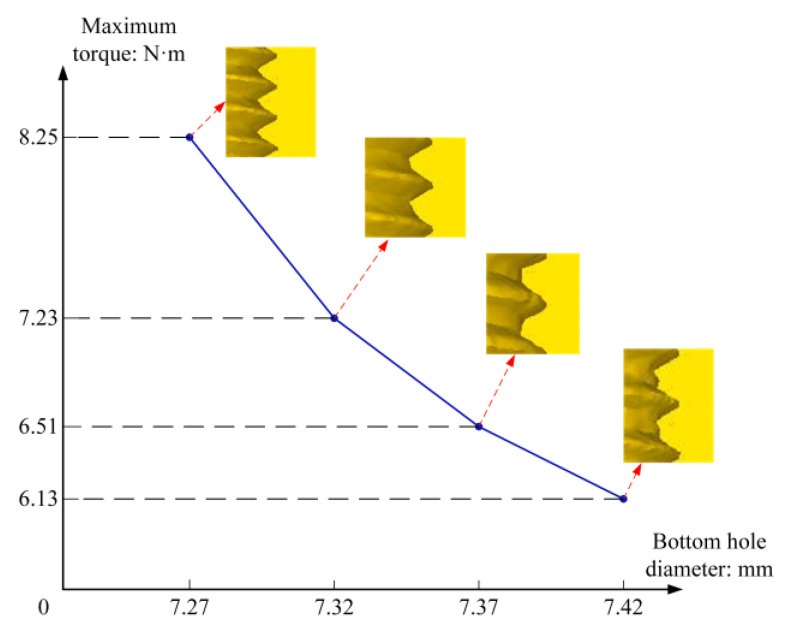
Influence of bottom hole diameter on extrusion effect.

**Figure 6 materials-13-03960-f006:**
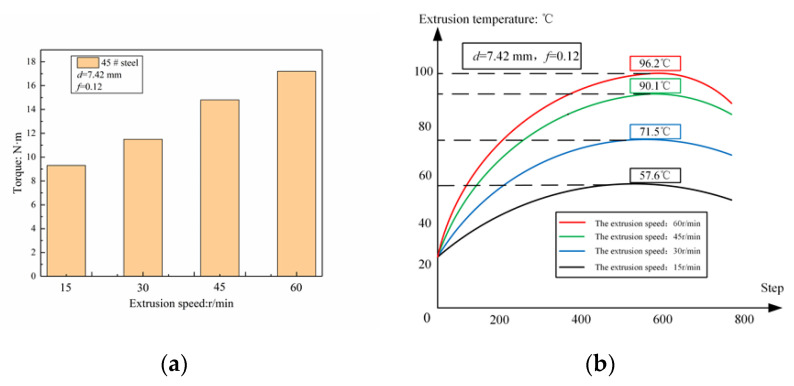
The influence of extrusion speed on torque (**a**) and the influence on temperature (**b**).

**Figure 7 materials-13-03960-f007:**
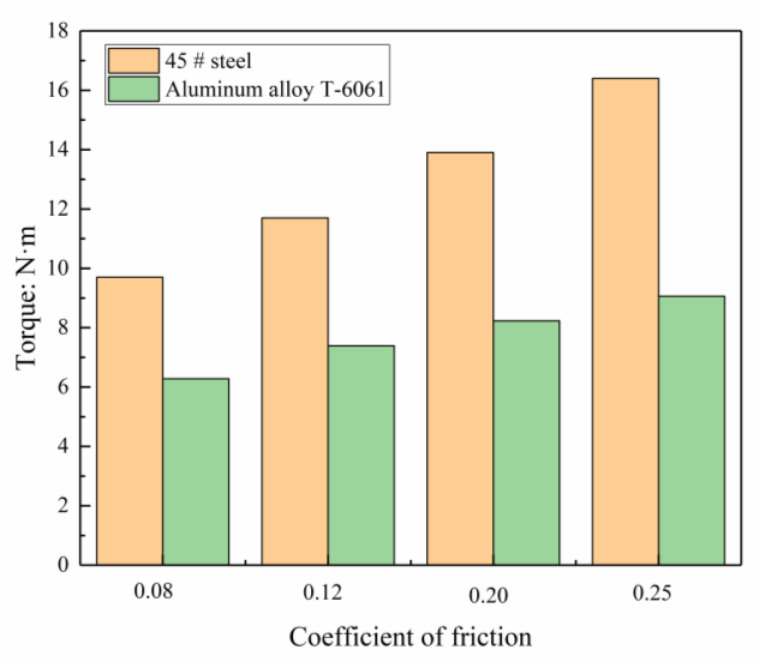
The influence of cooling lubrication on torque.

**Figure 8 materials-13-03960-f008:**
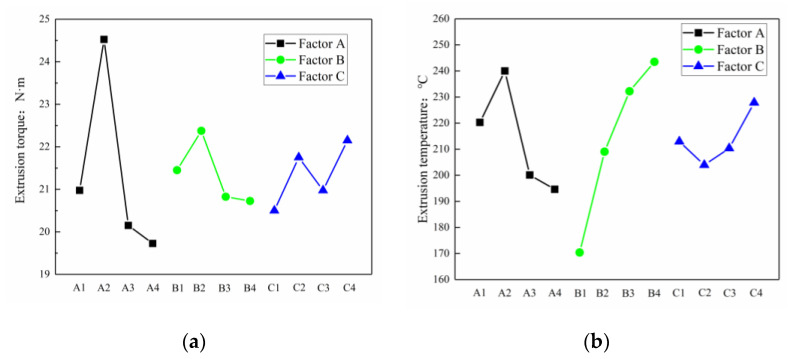
The influence of various factors on the extrusion torque (**a**) and the influence on the extrusion temperature (**b**).

**Figure 9 materials-13-03960-f009:**
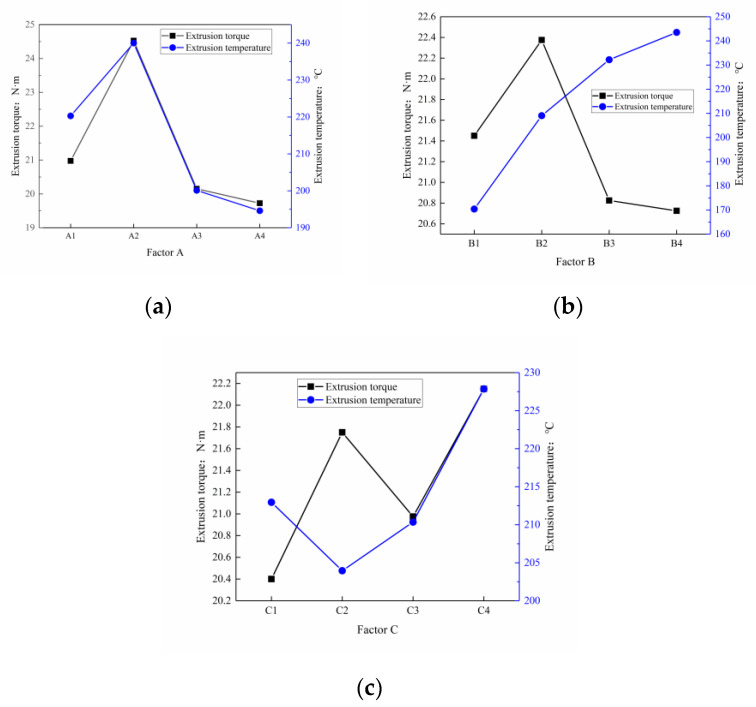
The influence of various factors on the test index: A factor (**a**), B factor (**b**), and C factor(**c**).

**Figure 10 materials-13-03960-f010:**
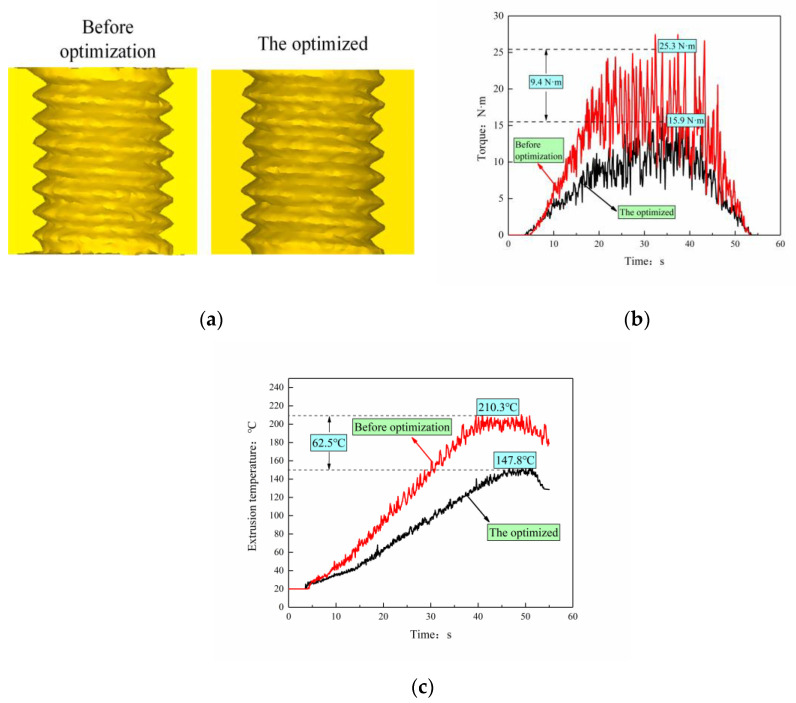
Comparison before and after optimization: (**a**) Extrusion profile, (**b**) Extrusion torque, (**c**) Extrusion temperature.

**Figure 11 materials-13-03960-f011:**
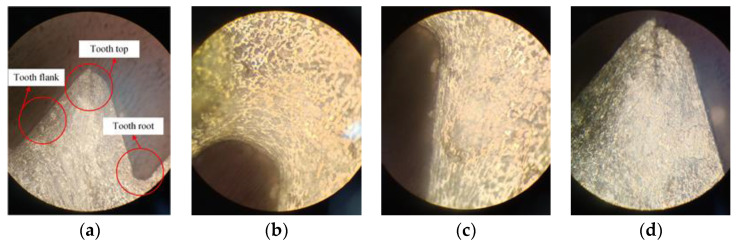
Extrusion profile measurement: (**a**) Three measurement positions, (**b**) Tooth root, (**c**) Flank, and (**d**) Tooth top.

**Figure 12 materials-13-03960-f012:**
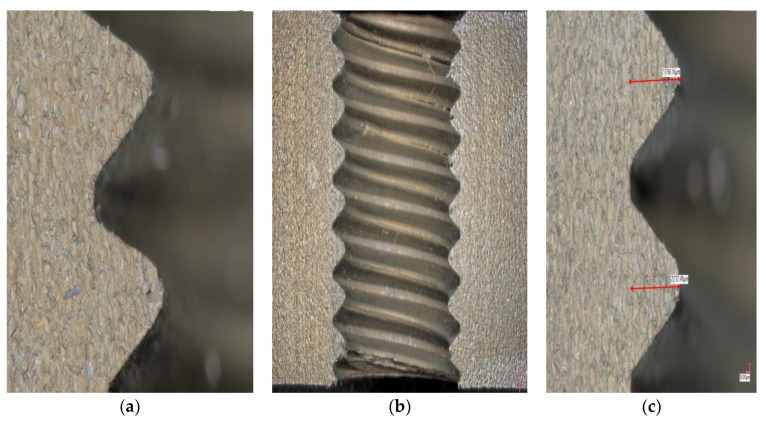
Contrast of tooth shape before and after optimization of process parameters (**a**) Tooth shape before optimization, (**b**) Optimized extrusion effect, (**c**) Optimized tooth shape.

**Table 1 materials-13-03960-t001:** Extrusion conditions of 45 # steel.

Parameters	Values
Φ (Blank diameter)/mm	30
d (Bottom diameter)/mm	7.30
n (Extrusion speed)/r·min^−1^	30
f (Friction factor)	0.08

**Table 2 materials-13-03960-t002:** Factor level.

Level	Diameter of Bottom Hole (mm): A	Extrusion Speed (r/min): B	Friction Factor: C
1	7.25	30	0.08
2	7.30	60	0.12
3	7.35	90	0.20
4	7.40	120	0.25

**Table 3 materials-13-03960-t003:** Simulation results.

Number	A	B	C	Extrusion Torque (N·m)	Extrusion Temperature (°C)
1	1	1	1	19.1	166.4
2	1	2	2	22.7	203.6
3	1	3	3	19.7	232.9
4	1	4	4	22.4	278.2
5	2	1	2	25.6	194.3
6	2	2	1	25	238.9
7	2	3	4	23.7	269.9
8	2	4	3	23.8	257.4
9	3	1	3	19.8	161.3
10	3	2	4	21.2	203.8
11	3	3	1	20.4	221.7
12	3	4	2	19.2	213.6
13	4	1	4	21.3	159.5
14	4	2	3	20.6	189.8
15	4	3	2	19.5	204.3
16	4	4	1	17.5	224.8

**Table 4 materials-13-03960-t004:** Range analysis.

Test Index	Factor	$\bar{K_{1}}$	$\bar{K_{2}}$	$\bar{K_{3}}$	$\bar{K_{4}}$	Range Value R
Extrusion torque (N·m)	A	20.975	24.525	20.150	19.725	4.800
	B	21.450	22.375	20.825	20.725	1.650
	C	20.400	21.750	20.975	22.150	1.750
Extrusion temperature (°C)	A	220.275	240.000	200.100	194.600	45.400
	B	170.375	209.025	232.200	243.500	73.125
	C	212.950	203.950	210.350	227.8500	23.900

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
