# Peer review of "Numerical Simulation and Process Optimization of Internal Thread Cold Extrusion Process"

_materials, 2020, doi:10.3390/ma13183960_

Round 1
Reviewer 1 Report
In the introduction of the article, the authors presented a study of literature which is connected to the main subject of the paper.
Authors highlight that many of the used dies in industry machines are not properly designed. This problem influences the efficiency of industrial processes which also was the subproblem of many other articles. In my opinion, 17 references are not much in the case of Journal Materials. I highly recommended preparing a little bit wider literature study focused on die design for example doi.org/10.3390/ma13153317 .
During the pre-review process was noted below pointed signification disadvantages:
- The quality of all figures are very low. for example text on fig. 4 are unreadable;
- In figures 10 &12, the zoom-in of the analyzed model is not enough for understanding the results of the optimization process.
- Results of the optimization process are not enough presented, I suggest to highlight the effects of work;
- Faults in the symbols in the text, for example, line 244 and symbols K1, K2, K3, and K4;
- Faults in references - double number o each position;
Author Response
Dear Editor and Reviewers,
We would like to thank the materials for giving us the opportunity to revise our manuscript.
We do appreciate your effort and the reviewers’ close review on our previous manuscript. Thank you for your constructive suggestions and comments that help us improve the paper both in wording and in technical aspect. The revision of the paper is attached. We revised the paper according to your comments and suggestions. Efforts were also made significantly to improve the English, such as reorganize the sentences, replace inappropriate phrase so on and so forth.
We appreciate your taking the time to review my new manuscript. Again, thank you for your consideration.
Sincerely yours,
Guangpeng Zhang
School of Mechanical and Precision Instrumental Engineering
Xi’an University of Technology
No.5, Jinhua Road,
Xi’an, Shaanxi Province 710048
- R. China
E-mail: gpzhang@xaut.edu.cn
Tel.: +86 13991368732
We thank the reviewers for their careful read and thoughtful comments on the previous draft. Those comments are all valuable and very helpful for revising and improving our paper. We have carefully taken their comments into consideration in preparing our revision and we hope the revised version could meet with approval.
Our responses to the reviewer’s comments are as flowing:
Reviewer #1:
Point 1. Results of the optimization process are not enough presented, I suggest to highlight the effects of work.
Response: Thank you for the comment. In the revised manuscript, we have explained the optimized results in detail. We have included the results of tooth height and density after the extrusion process. At the same time, we have included the comparison of tooth profile before and after optimization.
The effects of the optimization process are significant. Through the simulation tests, the tooth height is 0.839 mm before optimization process and is 0.823 mm after optimization, the ratios of formed tooth height and required tooth height are 77.47% and 76.04% respectively. The simulation results are also verified with experimental tests. The experiment results are shown in Figure xx. The tooth height values are 0.852 mm before optimization and 0.840 mm after optimization, the ratios of formed tooth height and required tooth height are 78.68% and 77.62% respectively. The simulations results are in good agreement with the experiment results.
Figure xx Formed tooth profile before optimization (formed tooth profile after optimization is shown in Figure 12 of the revised manuscript)
Thank you for your suggestion, I will further modify the introduction。

Reviewer 2 Report
The authors combined numerical simulation and orthogonal experiments to analyze the influence of the bottom hole diameter, extrusion speed and friction factor on the extrusion torque and extrusion temperature during internal thread extrusion forming. The authors found that the optimized extruded thread had good results in terms of contour, tooth pitch/shape/flatness, and the maximum extrusion torque and temperature was reduced by 37.15% and 29.72%, respectively.
The paper is interesting, but some modifications should be tackled before publication:
In the first paragraph of the introduction section, the authors refer to “the country” or “my country”. Please define the country and try to use impersonal writing.
In line 53, include the meaning of BP (definition the first time the acronym is mentioned).
In the last paragraph of section 1, there are several “,” followed by capital letters. Please review this, as it leads to misunderstandings.
Please improve the quality of all the figures.
Equation 3 is not shown completely.
In line 105, please include the “°” symbol to indicate the units of the angle (60°).
In line 121, the number of grids (100,000) refers to the number of nodes?
Table 1: Is “Diameter of bottom hole” the internal diameter or the bottom diameter in the external edge of the chamfer (Figure 4b)?
Please provide more information about the finite element model: how did constrained the tap? How did you obtain the responses (maximum torque/temperature)? Element type? Try to give all the needed information to reproduce this simulation.
Line 145: Improve the link between sentences.
Line 181: I would suggest using another term rather than “machined surface”.
English writing can be improved (some sentences need revision).
Line 212: I would recommend to change the term “machine speed”
Section 4.2.2.: Please improve the explanation and details of the range analysis (meaning of K). Also use Kj instead of Ki, or change the index.
Table 4 is not clear, at least in the PDF version. The factors and corresponding level of each set is not clear.
Line 224: Is “Range analysis” a title?
Table 3 is not clear (there are parts of the same numbers in several lines).
Line 230: Use “,” instead of “.”.
In figure 8, does the x axis represent levels 1-4? Please add this information in the caption of x axis.
Line 236: the best combination when minimizing the extrusion temperature should be A4B1C2.
Line 247: Why do you state that the best level for factors B and C are B1 and C1? Are you considering the same weight for both responses (torque and temperature)? Please clarify how you objectively chose that levels as the optimums.
Line 248: consider changing the term “machine speed”.
Section 5.1.: Please include the factors used to obtain the profile “before optimization”, so that the readers can compare and know the parameters used in each case.
Line 261: “It can be seen that there is no significant change in the height of the internal thread before and after optimization”. This is not quantified or justified. Please clarify this. In fact, the best parameters configuration could be obvious (the lowest torque and temperature should take place with the highest hole diameter, the lowest extrusion speed and the lowest friction factor). Therefore, the key is to check if with those parameters, the profile is ok, or if the forming time is not excessive. I checked that this profile analysis was carried out with the real test.
In that sense, this is not a pure optimization, but a selection of the best parameters tested (maybe this should be clarified in the abstract).
In figure 11, please include another picture indicating the root, flank and top form where the pictures were taken, so that the current pictures can be interpreted more easily.
In the conclusions section, the authors should explain how this research could help in the optimization of parameters, since the selected optimal parameters were almost obvious if there is not any constraint in the optimization (time, surface quality, etc.).
Author Response
Dear Editor and Reviewers,
We would like to thank the materials for giving us the opportunity to revise our manuscript.
We do appreciate your effort and the reviewers’ close review on our previous manuscript. Thank you for your constructive suggestions and comments that help us improve the paper both in wording and in technical aspect. The revision of the paper is attached. We revised the paper according to your comments and suggestions. Efforts were also made significantly to improve the English, such as reorganize the sentences, replace inappropriate phrase so on and so forth.
We appreciate your taking the time to review my new manuscript. Again, thank you for your consideration.
Sincerely yours,
Guangpeng Zhang
School of Mechanical and Precision Instrumental Engineering
Xi’an University of Technology
No.5, Jinhua Road,
Xi’an, Shaanxi Province 710048
- R. China
E-mail: gpzhang@xaut.edu.cn
Tel.: +86 13991368732
We thank the reviewers for their careful read and thoughtful comments on the previous draft. Those comments are all valuable and very helpful for revising and improving our paper. We have carefully taken their comments into consideration in preparing our revision and we hope the revised version could meet with approval.
Our responses to the reviewer’s comments are as flowing:
Reviewer #2
Point 1: In line 121, the number of grids (100,000) refers to the number of nodes?
Response: Thank you for the comment. Yes, it is the number of nodes. We have modified the expression to "the number of nodes".
Point 2: Table 1: Is “Diameter of bottom hole” the internal diameter or the bottom diameter in the external edge of the chamfer (Figure 4b)?
Response 2: Thank you for the comment. It is the "bottom diameter".
Point 3: Please provide more information about the finite element model: how did constrained the tap? How did you obtain the responses (maximum torque/temperature)? Element type? Try to give all the needed information to reproduce this simulation.
Response 3: he movement parameters of the extrusion tap are set according to the extrusion speed, and each lead is fed along the axial direction while rotating by 2π rad. fter the finite element simulation is over, the "torque" function can be selected in the "load displacement diagram" module built in DEFORM-3D software to obtain the change trend of the extrusion torque during the extrusion process. In the "Summary" module, select "Heat Transfer Mode-Temperature" to obtain the temperature trend during the extrusion process.
Point 4: Line 145: Improve the link between sentences. 第145行:改进句子之间的联系。
Response 4: Thank you for the comment. We have modified these sentences in the revised manuscript.
Point 5: English writing can be improved (some sentences need revision).
Response 5:
Thank you for the comment. We have check the English carefully and modifed the related content to avoid the grammatical errors.
Point 6: Section 4.2.2.: Please improve the explanation and details of the range analysis (meaning of K). Also use Kj instead of Ki, or change the index.
Response 6: Thank you for the comment. Ki represents the sum of the corresponding test results when the level number is i in any column i=1,2,3,4). represents the average of Ki.
Point 7:Line 224: Is “Range analysis” a title?
Response 7: Yes, "1)" is omitted at the beginning of the sentence.
Point 8:Line 230: Use “,” instead of “.”.
Response 8: Are you referring to the data in the table? For example, the "." in "20.975", if it is, in the original version of the article, due to my typesetting mistakes, you mistakenly believed that data like "20.975" was separated, in fact, "." is a decimal point.
Point 9: Line 236: the best combination when minimizing the extrusion temperature should be A4B1C2.
Response 9: Thank you for the comment. We have modified the mistake in the revised manuscript.
Point 10: Line 247: Why do you state that the best level for factors B and C are B1 and C1? Are you considering the same weight for both responses (torque and temperature)? Please clarify how you objectively chose that levels as the optimums.
Response 10: Thank you for the comment. Through range analysis, the order of the factors affecting the extrusion torque is A>C>B for extrusion torque. That is, the factor B has the greatest influence on extrusion temperature and has the lowest influence on extrusion torque. Therefore, B1 is selected as the optimized parameters. The factor C has the greater influence on extrusion torque than extrusion temperature. A small friction factor is able to reduce the load torque of screw tap during extrusion process, and the extrusion torque is directly related to the service life of screw tap. Therefore, C1 is selected as the optimized parameters considering the influences of factor C on each indicator.
Point 11:Section 5.1.: Please include the factors used to obtain the profile “before optimization”, so that the readers can compare and know the parameters used in each case.
Response 11: The process parameters before optimization are: extrusion speed 60r/min, bottom hole diameter is 7.35 mm, friction coefficient is 0.20.
Point 12: Line 261: “It can be seen that there is no significant change in the height of the internal thread before and after optimization”. This is not quantified or justified. Please clarify this. In fact, the best parameters configuration could be obvious (the lowest torque and temperature should take place with the highest hole diameter, the lowest extrusion speed and the lowest friction factor). Therefore, the key is to check if with those parameters, the profile is ok, or if the forming time is not excessive. I checked that this profile analysis was carried out with the real test.
Response 12: Through the simulation tests based on DEFORM-3D (Figure 10a), the tooth height is 0.839 mm before optimization process and is 0.823 mm after optimization, the ratios of formed tooth height and required tooth height are 77.47% and 76.04% respectively. The tooth height is measured with the "Ruler tool" of DEFORM-3D. According to the simulation results, the tooth height does not change so much before and after the optimization. However, the extrusion torque and extrusion temperature can be obviously reduced using the optimized process parameters.
Through the extrusion experiments with the optimized process parameters, the results show that the formed tooth profile is full. For the workpiece with the thick of 10 mm, the whole extrusion process takes less than 80 seconds.
Point 13: In that sense, this is not a pure optimization, but a selection of the best parameters tested (maybe this should be clarified in the abstract).
Response 13: Thank you for the comment. We have modified the expression in Abstract.
Point 14: In figure 11, please include another picture indicating the root, flank and top form where the pictures were taken, so that the current pictures can be interpreted more easily.
Response 14: Thank you for the comment. We have included a figure to explain the positions of microstructure pictures.
Point 15: In the conclusions section, the authors should explain how this research could help in the optimization of parameters, since the selected optimal parameters were almost obvious if there is not any constraint in the optimization (time, surface quality, etc.).
Response 15: Thank you for the comment. We have explained the constraint of optimization in the revised manuscript. The constraint of the optimization is: For the internal thread with M8×1.25 mm, the optimization objectives of extrusion process are low extrusion torque and low temperature in extrusion area. In this work, we have optimized the process parameters based on multi-object and multi-parameter. In addition, the optimization method in this manuscript is also suitable for the extrusion parameter selection of other size.
Round 2
Reviewer 1 Report
After revision paper has been improved by authors.